# Small disulfide loops in peptide hormones mediate self-aggregation and secretory granule sorting

Jennifer Reck, Nicole Beuret, Erhan Demirci, Cristina Prescianotto-Baschong, Martin Spiess

**Unlike constitutively secreted proteins, peptide hormones are stored in densely packed secretory granules, before regulated release upon stimulation. Secretory granules are formed at the TGN by self-aggregation of prohormones as functional amyloids. The nonapeptide hormone vasopressin, which forms a small disulfide loop, was shown to be responsible for granule formation of its precursor in the TGN as well as for toxic fibrillar aggregation of unfolded mutants in the ER. Several other hormone precursors also contain similar small disulfide loops suggesting their function as a general device to mediate aggregation for granule sorting. To test this hypothesis, we studied the capacity of small disulfide loops of different hormone precursors to mediate aggregation in the ER and the TGN. They indeed induced ER aggregation in Neuro-2a and COS-1 cells. Fused to a constitutively secreted reporter protein, they also promoted sorting into secretory granules, enhanced stimulated secretion, and increased Lubrol insolubility in AtT20 cells. These results support the hypothesis that small disulfide loops act as novel signals for sorting into secretory granules by self-aggregation.**

## Introduction

The constitutive secretory pathway continuously transports newly synthesized proteins to the plasma membrane and the extracellular space in all eukaryotic cells. Endocrine cells, in addition, possess a regulated pathway where proteins are packaged into secretory granules for storage, until release is triggered by external stimulation (Dannies, 1999; Kim et al, 2006). Both pathways share the early steps from the ER to the Golgi and diverge only at the TGN (Guo et al, 2014; Ramazanov et al, 2021). Here, the contents of secretory granules are concentrated in a densely packed form leading to the dense-core appearance in electron microscopy. Regulated cargo includes protein hormones and granins, as well as processing enzymes such as prohormone convertases and carboxypeptidases (Taupenot et al, 2003; Seidah & Prat, 2012). Association with cholesterol-rich lipid microdomains leads to

formation of immature secretory granules without an obvious cytosolic coat. They mature by removal of much of contaminating proteins by so called constitutive-like secretion involving AP-1/clathrin coats (Dittie et al, 1996; Arvan & Castle, 1998), by further acidification from pH ~6.3 in the TGN to pH 5.0–5.5 in mature granules (Urbé et al, 1997), by homotypic fusion, and by condensation and removal of water (Kim et al, 2006).

So far, no generally accepted cargo receptors have been identified for the sorting of hormone proteins and granins into secretory granules. Amphipathic helices interacting with the granule membrane have been found to be important for the incorporation of prohormone convertases and carboxypeptidase E (e.g., Dikeakos et al, 2009). Di-basic processing sites and acidic motifs in prohormones were shown to promote granule sorting, possibly by interaction with the convertases (e.g., Brechler et al, 1996) and with carboxypeptidase E (Lou et al, 2005), respectively.

Importantly, a major role in granule formation—collecting and concentrating cargo and displacing other proteins—is played by self-aggregation of regulated cargo in the TGN (Chanat & Huttner, 1991; Arvan & Castle, 1998; Kim et al, 2006; Dikeakos & Reudelhuber, 2007; Dannies, 2012). A direct correlation between the ability to aggregate in vitro and to be sorted to secretory granules has first been reported for the precursor of atrial natriuretic peptide (Canaff et al, 1996) and chromogranin A (CgA) (Jain et al, 2002). Indeed, expression of prohormones and granins was shown to be sufficient to produce dense granule-like structures in fibroblasts and other cells lacking endocrine-specific machinery (Kim et al, 2001; Huh et al, 2003; Beuret et al, 2004). It is thus the specific environment of the TGN—reduced pH, high concentrations of $Ca^{2+}$ and possibly other divalent cations, and the presence of glycosaminoglycans (Kolset et al, 2004; Dannies, 2012)—that triggers aggregation of regulated cargo proteins.

Because granule contents therefore are reversible aggregates of folded proteins, it came as a surprise when Riek and colleagues (Maji et al, 2009) proposed that secretory granules of pituitary peptide hormones constitute functional amyloids. Typically, amyloids are fibrillar insoluble aggregates made of $\beta$-sheets with repetitive $\beta$-strands perpendicular to the fiber axis (cross-$\beta$ structure). In living cells, amyloids have long been associated exclusively with disease. In recent years, however, functional amyloids have been

---

Biozentrum, University of Basel, Basel, Switzerland

Correspondence: martin.spiess@unibas.ch

discovered that perform a number of physiological roles (Otzen & Riek, 2019).

Maji et al (2009) showed that a large number among 42 tested peptide hormones produced fibrillar aggregates in vitro under TGN-like conditions with reduced pH and glycosaminoglycans. These in vitro aggregates stained positive with amyloid dyes and in several cases produced the typical X-ray diffractions at 4.7 and ~10 Å. Material from purified secretory granules of AtT20 cells or rat pituitary stained positive for the amyloid-specific antibody OC and produced fluorescence with Thioflavin T. Most significantly, mouse pituitary tissue sections revealed costaining of secretory granules positive for prolactin, adrenocorticotropic hormone, growth hormone, oxytocin, or vasopressin with Thioflavin S.

The notion that peptide hormones form amyloids at the TGN may explain that quite a few of them are also known to cause amyloid diseases (Westermark, 2005). Examples are amylin (islet amyloid polypeptide, IAPP) associated with type 2 diabetes mellitus, procalcitonin with medullary carcinoma of the thyroid, atrial natriuretic factor with atrial amyloidosis, and provasopressin causing familial neurohypophyseal diabetes insipidus. Vasopressin is an antidiuretic hormone released from the posterior pituitary and regulates water resorption in the kidney. Diabetes insipidus is a dominant disease caused by folding-deficient mutants of provasopressin that are retained in the ER and produce fibrillar aggregates (Birk et al, 2009). Upon coexpression of wild-type and mutant precursors, the wild-type protein is also retained by coaggregation with the mutant (Ito et al, 1999). In mice, vasopressinergic neurons expressing a mutant transgene were shown to contain large aggregates and to be eliminated by autophagy-associated cell death (Hagiwara et al, 2014), replicating the human situation, where vasopressinergic neurons were lost in diabetes insipidus patients (Green et al, 1967).

Pre-provasopressin (illustrated in Fig 1, top) consists of a signal sequence for ER targeting, the nonapeptide sequence of the final vasopressin hormone, the 93-amino acid neurophysin II (NPII) domain, and a glycopeptide of 39 residues of unknown function. Vasopressin, NPII, and glycopeptide are proteolytically cleaved in secretory granules. By systematic mutagenesis, vasopressin and the glycopeptide were identified to be each sufficient to cause aggregation in the ER of mutant NPII (Beuret et al, 2017). Deletion of the glycopeptide and mutation of vasopressin inactivated ER

aggregation of a folding-deficient NPII and also strongly reduced sorting into granules and regulated secretion in AtT20 cells (Beuret et al, 2017). These results supported the aggregation model of granule biogenesis and suggested that the sequences that had evolved to mediate aggregation of the precursor at the TGN are responsible for pathological aggregation of mutant provasopressin in the ER to cause dominant disease.

Vasopressin is particularly interesting because it contains a disulfide bond between Cys-1 and Cys-6, forming a small loop structure. Vasopressin was one of the pituitary hormones tested positive by Maji et al (2009) for in vitro amyloid formation and for Thioflavin T and Congo Red binding as an oxidized nonapeptide. In X-ray diffraction, the aggregates produced extra reflections at 4.4 Å suggesting unusual and cross $\beta$-sheet atypical properties. This may be due to the small ring structure that precludes an extended conformation as required for $\beta$-strands. Interestingly, small disulfide loops (CC loops) are present in a number of peptide or protein hormones (Table 1), frequently at the very N- or C terminus of the (precursor) protein or close to processing sites and thus potentially exposed. In the case of pro-opiomelanocortin (POMC), an N-terminal 13-residue CC loop had been shown to be necessary and sufficient for granule sorting (Tam et al, 1993; Cool et al, 1995). These considerations led to the hypothesis that CC loops may generally act as aggregation devices for sorting into secretory granules at the TGN.

Here, we tested the CC loops of four additional peptide hormones—of amylin, calcitonin, prorenin, and prolactin—for their ability to aggregate at the ER level like vasopressin and to sort a constitutive reporter protein into granules at the TGN. Our results support the hypothesis that CC loops in general contribute to granule sorting via self-aggregation.

# Results

## CC loops mediate ER aggregation to different extents

The starting point to study the ability of different CC loops to mediate aggregation was a truncated version of pre-provasopressin, CCv-NPΔ

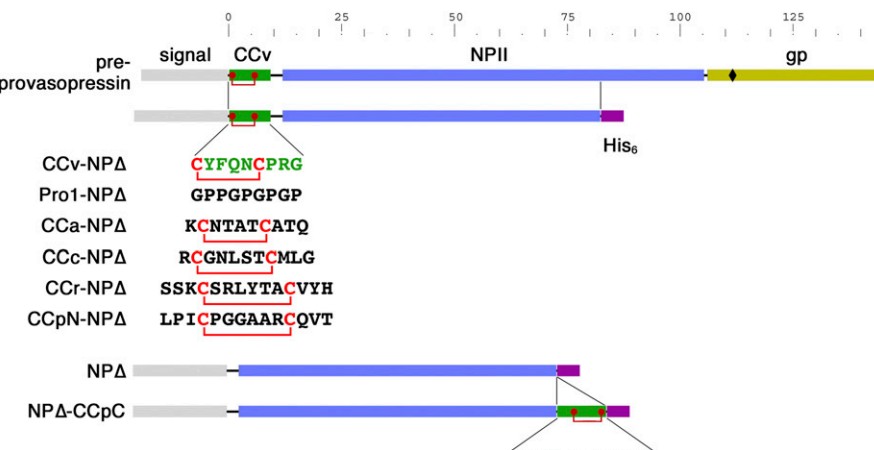

**Figure 1. CC loop reporter for ER aggregation.**
Schematic representation of pre-provasopressin, which is composed of a signal sequence, CCv (the nonapeptide vasopressin with its disulfide bond in red), neurophysin II (NPII), and the glycopeptide (gp; glycosylation site indicated as a diamond), and the truncated NPΔ constructs. For the listed constructs, the sequence of CCv was replaced by that of Pro1, by the CC loop sequences of amylin (CCa), calcitonin (CCc), renin (CCr), or the N-terminal CC loop of prolactin (CCpN). The C-terminal CC loop of prolactin (CCpC) was inserted C-terminally between the NPII sequence and the hexahistidine (His6) tag. The scale indicates the number of the amino acids in provasopressin.

**Table 1. CC loops in peptide hormone precursors.**

| Hormone | Source | Loop size | N-Terminal | Internal | C-Terminal |
|---|---|---|---|---|---|
| Vasopressin | pp | 6 | ↑**CYFQNC**PRG↓ | | |
| Oxytocin | pp | 6 | ↑**CYIQNC**PLG↓ | | |
| POMC | ap | 13 | ↑WCLESSQ**CQDLTTESNLLEC**IRAC... | | |
| Prolactin | ap | 8+9 | ↑LPI**CPGGAARC**QVT... | | ...LLN**CRIIYNNNC**· |
| Growth hormone | ap | 9 | | | ...VMK**CRRFAESSC**AF· |
| Somatostatin | ht | 12 | | | ...↓AG**CKNFFWKTFTSC**· |
| Cortistatin | cc | 12 | | | ...RMP**CRNFFWKTFSSC**K· |
| Urotensin II | mn | 6 | | | ↓ETPD**CFWKYC**V· |
| Urotensin IIB | mn | 6 | | | ↓A**CFWKYC**V· |
| Amylin | β | 6 | | ↓K**CNTATC**ATQ... | |
| Calcitonin | th | 7 | | ↓**CGNLSTC**MLG... | |
| Prorenin | jg | 8 | | ...SSK**CSRLYTAC**VYH... | |
| Osteocalcin | ob | 7 | | ...REV**CELNPDC**DEL... | |

POMC, pro-opiomelanocortin, pp, posterior pituitary; ap, anterior pituitary; ht, hypothalamus; cc, cerebral cortex; mn, motorneurons; jg, juxtaglomerular cells (kidney); β, β cells (pancreas); th, thyroid; ob, osteoblasts; ↑ signal cleavage site; ↓ convertase processing site; · C terminus; the CC loop sequences are in bold.

(Fig 1; called 1–75 in Beuret et al [2017]) that cannot fold, is retained in the ER, and produces fibrillar aggregates. It is composed of a signal sequence, the vasopressin hormone CCv, and a C-terminally truncated form of neurophysin II (NPΔ) fused to a hexahistidine (His₆) tag. For the negative control Pro1-NPΔ, CCv was replaced by the proline/glycine repeat sequence Pro1 that was previously shown to essentially abolish aggregate formation (Birk et al, 2009; Beuret et al, 2017), as well as NPΔ alone without N-terminal extension. Similarly, to test whether CC loops of other protein hormones are able to cause aggregation, CCv was replaced by the CC loops from amylin (CCa), calcitonin (CCc), renin (CCr), or by the N-terminal CC loop of prolactin (CCpN). The C-terminal CC loop of prolactin (CCpC) was inserted C-terminally of the NPΔ sequence (Fig 1).

These constructs were transiently transfected into COS-1 fibroblasts and Neuro-2a neuroblastoma cells. COS-1 cells are large and easy to analyze for aggregations by immunofluorescence microscopy, whereas Neuro-2a cells, although smaller and more difficult to analyze for aggregates by light microscopy, produced nice fibrillar aggregates detectable by electron microscopy for provasopressin mutants (Birk et al, 2009). Originally, Neuro-2a cells were used to study the behaviour of pathogenic diabetes insipidus mutants of provasopressin in a neuroendocrine cell type to reflect the situation in vasopressinergic neurons. For the analysis of ER aggregation, it does not matter that Neuro-2a but not COS-1 cells have a regulated secretory pathway because the NPΔ constructs cannot reach the TGN, where regulated and constitutive pathways separate.

In COS-1 cells (Fig 2A), Pro1-NPΔ and NPΔ were found evenly distributed in the ER network in most of the expressing cells. In contrast, CCv-NPΔ produced clear accumulations in most cells, confirming ER aggregation as previously observed (Birk et al, 2009; Beuret et al, 2017). The constructs containing CCa, CCpN, and to a lesser degree CCr also generated rather fine aggregates similar to those of CCv. CCpC-NPΔ frequently produced rather large structures

in a smaller number of expressing cells. Least efficient was CCc-NPΔ. In comparison, Neuro-2a cells displayed visible aggregates only upon expression of CCv, CCpN, and CCa (Fig 2B).

For quantitation, expressing cells were manually scored by immunofluorescence microscopy for aggregations. It confirmed the ability of all constructs to produce an increase of cells with aggregates above background in COS-1 cells (Fig 2C), although not statistically significant for CCc (*P* = 0.08). The highest aggregation propensity was found for CCpN, CCv, and CCa. In Neuro-2a cells, aggregation of CCpN and CCv was equally high, whereas it was reduced for CCa and even not statistically significant for CCr, CCpC, and again for CCc (Fig 2D).

This analysis thus shows differences in aggregation propensity between the different CC loops and between the expressing cell lines. The latter might be due to expression levels, which are likely to be higher in COS-1 cells where plasmids with SV40 origins—as is the case for the expression plasmid pcDNA3 we used—are amplified. Alternative explanations are differences in chaperone levels and/or ERAD capacities of the cell lines. Overall, it appears that the Neuro-2a expression system is less permissive of aggregation of our constructs and thus more clearly displays the differences in aggregation efficiency of the CC loops tested.

To confirm the localization of the CC-NPΔ aggregates in the ER, we expressed myc-tagged protein disulfide isomerase (PDI) for immunofluorescence microscopy (Fig S1A and B). As expected, myc-PDI expressed alone in either COS-1 or Neuro-2a cells was observed in the typical network pattern of the ER. When co-expressed with Pro1-NPΔ or the different CC-NPΔ constructs, myc-PDI clearly colocalized with each of them confirming the localization of the proteins in the ER. Myc-PDI not only colocalized with the NPΔ fusion proteins in the normal ER reticulum, but also concentrated in the aggregates. This result was not surprising because the neurophysin fragment containing nine cysteines is a substrate of PDI. It should

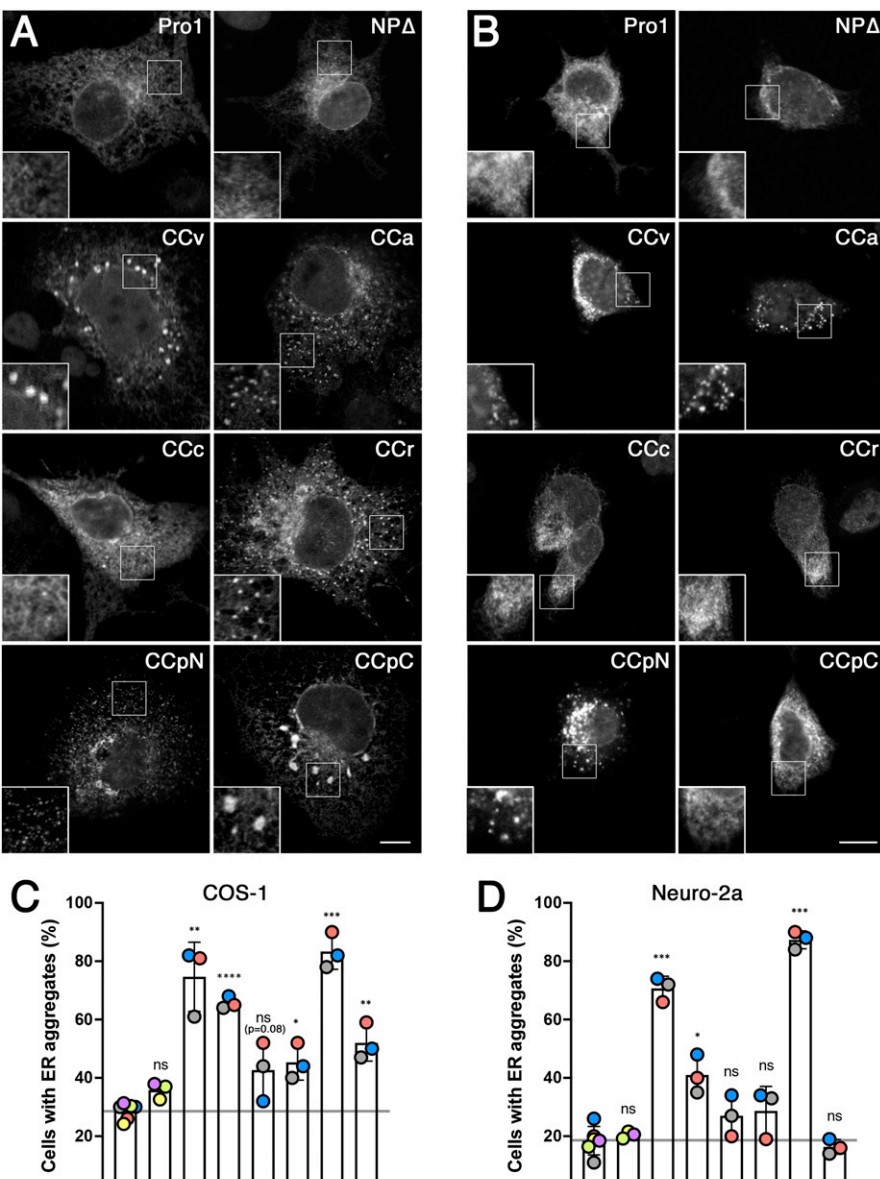

**Figure 2. ER aggregation of CC-NPΔ constructs in COS-1 and Neuro-2a cells.**

**(A, B)** COS-1 (A) and Neuro-2a cells (B) were transiently transfected with the NPΔ constructs listed in Fig 1, stained with anti-His$_6$ antibodies, and imaged by immunofluorescence microscopy. Insets show enlargements of indicated regions. Scale bars, 10 μm. **(C, D)** The number of cells with aggregates (i.e., displaying >3 apparent accumulations) were quantified from 100 to 150 expressing cells for each construct in each of three independent experiments for Pro1 and all CC-NPΔ constructs, and separately for Pro1 and NPΔ, colored by experiment. Statistical significance was calculated using the unpaired $t$ test. n = 3. ns, not significant; *$P \leq 0.05$; **$P \leq 0.01$; ***$P \leq 0.001$; ****$P \leq 0.0001$.

also be pointed out that the fraction of cells with visible aggregates was reduced upon coexpression with myc-PDI. Indeed, PDI overexpression is likely to increase the capacity of the cell to keep NPΔ constructs in solution and competent for degradation.

### Ultrastructure of CC-NPΔ aggregates

Folding-deficient mutants of provasopressin aggregating in the ER had previously been shown by electron microscopy to form amyloid-like fibrils (Beuret et al, 2017). To unambiguously identify structures containing CC-NPΔ proteins, we applied immunogold labeling using anti-His$_6$ antibodies and 10-nm colloidal gold conjugated secondary antibodies of transfected Neuro-2a cells (Fig 3).

The anti-His$_6$ antibodies allowed immunogold detection of the CC-NPΔ proteins in aggregates, but hardly at all in the normal ER, where apparently the proteins' concentrations were below the threshold of detection. Indeed, the gold staining was specific for concentrated protein: no gold staining was observed on other cellular structures or in mock-transfected cells, not even in Pro1-NPΔ transfected samples including secretory granules. Depending on the CC loop expressed, dense structures with different sizes and shapes were found to be labeled with gold. We observed gold-positive very round aggregates for CCv with a size of approximately 180–420 nm, roundish aggregates of similar size for CCpN, and frequently big, more elongated and irregularly shaped structures of up to ~2,400 nm for CCa. Surprisingly, even CCc, CCr, and CCpC, which did

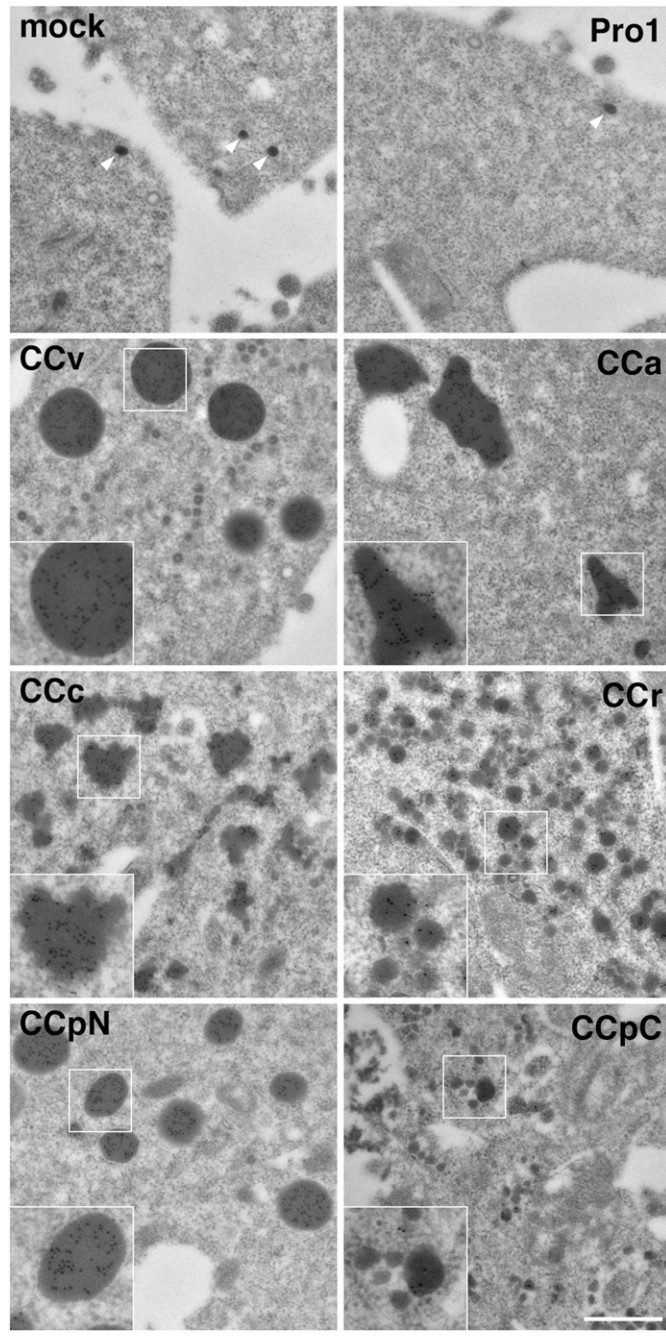

**Figure 3.  Ultrastructure of CC loop aggregates.**
Neuro-2a cells were mock-transfected or transfected with the indicated NPΔ fusion constructs, immunogold-stained using anti-His₆ antibodies and 10-nm gold-conjugated secondary antibodies, and imaged by electron microscopy. Representative gold-labeled aggregates of CC-NPΔ constructs are shown. The only dense structures found in mock- and Pro1-NPΔ–transfected cells were secretory granules of ~70 nm, devoid of immunogold (white arrowheads). Insets show twofold enlargements of the corresponding boxed areas. Scale bar, 500 nm.

not reveal significant aggregations by immunofluorescence, showed immunogold-positive dense structures with similar frequency. CCc aggregates produced smaller aggregations of heterogeneous shapes and sizes, whereas for CCr and CCpC, small structures of 60–80 and

30–150 nm, respectively, were produced. The discrepancy to the light microscopy experiments could be explained by the small sizes of the dense-structure, which certainly cannot be resolved by immunofluorescence and even in clusters might not be detectable. As to their internal morphology, the aggregates were very compact in all cases and no evidence of fibrillar substructure could be detected.

We conclude that besides vasopressin, CC loops of several other prohormones also have the capacity to mediate ER aggregation in different cell lines. The aggregates have different shapes and sizes depending on the CC loop involved. The ability of essentially all CC loop sequences to aggregate similarly to vasopressin (CCv) supports the hypothesis that they might also share this ability in the context of granule aggregation at the TGN.

### α1-protease inhibitor as a reporter for CC loop–mediated granule sorting

To directly investigate a role of CC loops in granule sorting, we fused CCv, CCa, CCc, CCr, and CCpN to α1-protease inhibitor (A1Pi, also called α1-antitrypsin) as a constitutively secreted reporter, tagged with a His₆ and a myc epitope tag (A1Pimyc). A1Pi has previously been used for a similar purpose to identify the N-terminal domain of CgB to mediate granule sorting (Krömer et al, 1998; Glombik et al, 1999). We produced one series of constructs with CC loops inserted between the signal sequence and the N-terminus of A1Pimyc (CC-A1Pimyc; Fig S2A). To potentially enhance an effect, we prepared additional constructs containing a second copy of the CC loop inserted between the C terminus of the A1Pi sequence and the tags (2xCC-A1Pimyc).

The A1Pimyc constructs were transfected into AtT20 mouse pituitary corticotropic cells. Stably expressing cells were selected and clonal cell lines isolated. Expression levels were assessed by metabolical labeling with [35S]methionine for 30 min, immunoprecipitation with anti-myc antibodies, SDS-gel electrophoresis, and autoradiography (Fig S2B). Cell lines of each construct were chosen for further analysis with very similar rates of synthesis mostly within ±20% of the control cell line expressing A1Pimyc without CC loop.

As expected, two forms of labeled A1Pimyc or CC-A1Pimyc constructs were produced, corresponding to the high-mannose ER form (lower band) and the complex glycosylated form (higher band; Fig S2B). The presence of the complex glycoform indicates that already within 30 min, significant amounts of newly synthesized A1Pi with and without CC loops had left the ER and reached the medial Golgi compartments, where glycan remodelling to the complex structures takes place. The presence of CC loops in the constructs thus did not interfere with A1Pi folding and did not cause ER retention themselves, a prerequisite to study sorting at the TGN.

### CC loops efficiently form their disulfide bond

In the context of the natural precursor, vasopressin/CCv is part of a folded structure with the disulfide bond protected in a binding pocket formed by neurophysin II (Chen et al, 1991; Wu et al, 2001). In our A1Pi fusion constructs, the CC loops are attached externally to the globular reporter. It is thus an important question, whether the cysteines in the added sequence are fully, partially, or not at all oxidized to a disulfide bond, when the protein exits the ER, reaches

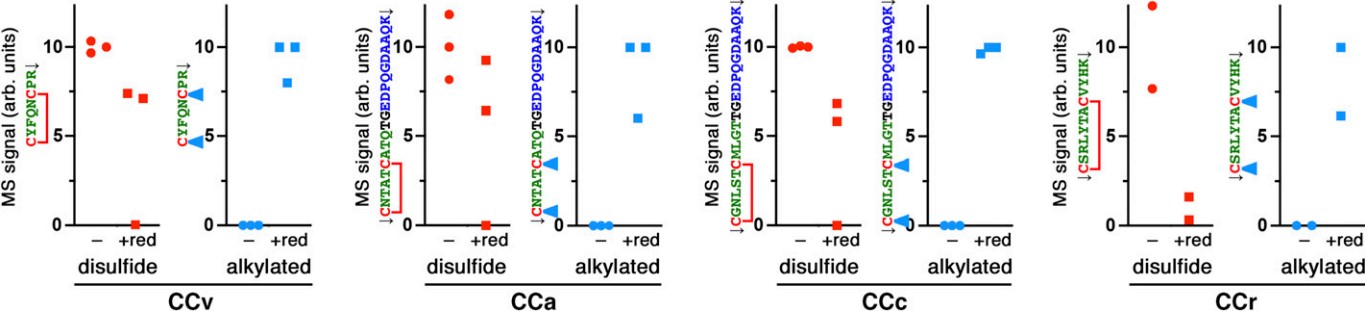

**Figure 4. Cysteines of CC loops are oxidized in secreted reporter fusion proteins.**
CC loop fusion proteins with A1Pimyc were immunoprecipitated from the media of producing AtT20 cell lines, reduced with TCEP (+red) or not (−), before alkylation of free cysteine thiols with chloroacetamide. Proteins were digested either with trypsin (CCv-, CCa-, and CCc-NPΔ) or Lys-C (CCr- and CCpN-NPΔ) and analyzed by mass spectrometry for the expected masses of the resulting CC-loop peptides with the cysteines either present as disulfides or in alkylated form. The signals (peak areas of the major ionic species) were normalized to the intensity of the peptide LQHLENELTHDIITK from within A1Pi and presented in arbitrary units (arb. units). Peptide sequences shown to the left of each graph: CC loop residues in green with red cysteines, either with a red disulfide bond or with two light blue triangles indicating alkylation, the N-terminal sequence of A1Pi in blue, and the linker sequence in black. Filled circles and squares indicate signals without and with reduction, respectively. Alkylated peptides were only detected after reduction. Disulfide forms of the peptides were detected without reduction and were decreased upon treatment with reducing agent (which was frequently rather inefficient). The peptides for CCpN-NPΔ (↓LPICPGGAARCQVTTGEDPQGDAAQK↓, disulfide bonded or alkylated) could not be detected. The results of two or three independent experiments are shown per construct.

the TGN, and eventually is secreted. To determine the oxidation state of the CC loops' disulfide bonds, media from the different CC-A1Pimyc expressing cells were collected and the constructs were immunoprecipitated and analyzed by mass spectrometry (Fig 4). For this, immunoprecipitates were each split into two samples, one to be reduced by tris(2-carboxyethyl)phosphine (TCEP), whereas the other remained untreated. Subsequently, both samples were incubated with chloroacetamide to carbamidomethylate (alkylate) the thiol groups of any free cysteines. After digestion with either trypsin or Lys-C protease, the resulting peptides that cover both cysteines of the CC loops either in the disulfide-bonded or the twice carbamidomethylated states were quantified by mass spectrometry.

The two forms of each of the four proteolytic fragments containing the disulfide loop of CCv, CCa, CCc, and CCr could be identified and quantified by mass spectrometry, but for unknown reasons not those of CCpN. They were only detected in the material of the cell line expressing the corresponding construct, demonstrating specificity. For all four peptides, the nonreduced sample exclusively contained the unmodified, disulfide-bonded version; no alkylated peptides were detectable, indicating that all secreted proteins contained the disulfide bond in their CC loop (Fig 4). Only after reduction (even though frequently incomplete) were the cysteines alkylated and the modified peptides readily detected by mass spectrometry.

## CC loops promote sorting into secretory granules

To analyze the ability of the CC loops to direct the reporter A1Pimyc into secretory granules, we analyzed their localization by immuno-fluorescence microscopy after costaining for our myc-tagged constructs and for the endogenous granule marker CgA. Quantitation of granule sorting requires normalization to a constant standard. Because there was some variation in staining between experiments and even between coverslips, we prepared an A1Pimyc cell line stably expressing cytosolic EBFP (enhanced blue fluorescent protein; A1Pimyc+EBFP; Fig 5A). Upon co-culturing every cell line together with these A1Pimyc+EBFP cells, blue fluorescence allowed to distinguish

the two cell types. The mean intensity of the anti-myc signal in the tips, as identified by CgA staining, was determined and normalized for that in co-cultured A1Pimyc+EBFP cells. Normalization to CgA could not be achieved because of an unexplained variation in CgA intensity in the tips depending on the protein expressed.

AtT20 cells have a spindle-shaped appearance with secretory granules collected in the tips. A1Pimyc was visible mainly in the ER and Golgi, but to some extent was also detected in the tips where CgA was accumulated (Fig 5A). It was not surprising to find constitutive cargo there because some missorting into immature secretory granules was expected, before being mostly removed via constitutive-like secretion (Feng & Arvan, 2003; Kim et al, 2006). However, a stronger signal in the tips was observed for cells expressing CCv-A1Pimyc and most other CC loop constructs (Fig 5A), thus confirming the ability of CC loops to promote granule sorting.

Quantitation confirmed that CCv increased storage of the reporter A1Pi in secretory granules by ~70%, whereas there was no difference in A1Pimyc localization between the A1Pimyc expressing cell lines with and without EBFP expression (Fig 5C). The other CC loop constructs accumulated to different extents in granules, CCc almost threefold, CCa and CCr similar to CCv, whereas there was a smaller increase by CCpN. Obviously, some CC loops, as CCc and CCr seem to have a huge impact on A1Pimyc sorting, whereas CCpN was not statistically significant ($P = 0.062$). The same experiment was also performed for the cell lines expressing the A1Pimyc constructs containing two copies of CC loops (Fig 5B and C). These constructs did not generally improve granule accumulation, but showed a positive effect in the case of CCr ($P = 0.003$).

The results show for four out of five CC loops, with the exception of CCpN, a significant increase in granule localization of the reporter protein.

## CC loop constructs are secreted from functional secretory granules

As an independent assay for granule sorting of CC loop constructs, we tested them for stimulated secretion. For this, the cells were

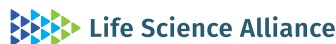

**Figure 5. CC loops promote A1Pi sorting into secretory granules.**
**(A, B)** Cell lines expressing A1Pimyc with or without EBFP, or A1Pimyc fusion proteins with one (A) or two CC loop sequences (B) were stained for myc and endogenous CgA. Arrowheads point out secretory granules in cellular tips in the control cell lines. Scale bars, 10 $\mu$m. **(C)** For quantitation, cell lines expressing A1Pimyc or the indicated CC loop constructs were grown mixed together with A1Pimyc+EBFP cells, stained, and imaged as in (A) and (B). Granules in tips as defined by CgA staining were quantified for the myc signal and normalized to that of A1Pimyc in A1Pimyc+EBFP cells on the same coverslip. 20 images with two to eight cells (~100 cells total) of each of them were analyzed per coverslip in three independent experiments. Statistical significance was determined using unpaired $t$ test. ns, nonsignificant; *$P \leq 0.05$; **$P \leq 0.01$; ***$P \leq 0.001$.

incubated either in medium lacking divalent cations to assess the resting secretion of the proteins mainly by the constitutive pathway or in stimulation medium containing barium chloride to induce the release of granule contents by increasing intracellular calcium (e.g., Bonnemaison et al [2014]).

As a positive control, we first analyzed stimulated secretion of the endogenous CgA from parental AtT20 cells. Resting and stimulating secretion media were collected after 1 h incubation and the aliquots analyzed by quantitative fluorescent immunoblotting (Fig 6A). Four forms of CgA were detected: unprocessed full-length CgA, as well as one small and two large processed fragments. Proteolytic processing is a sure indication of granule sorting. Accordingly, the processed bands were strongly increased in the medium upon stimulation, whereas the full-size protein remained the same (or was even slightly reduced), indicating constitutive secretion. The intensities of the processed bands in the non-stimulated sample were too low to reliably quantify a stimulation factor (the fold-increase of release into media with $BaCl_2$ versus without). The presence of a significant amount of non-stimulatable full-length CgA in the medium reveals the incompleteness of granule sorting for an endogenous regulated protein in AtT20 cells.

Because our A1Pimyc constructs are not processed in secretory granules, we cannot distinguish the pathways taken by the molecules based on their electrophoretic mobility. For comparison, we quantified the change in total secreted protein also for CgA—normalizing for VHH-mCherry nanobodies added to the media as a loading control and for the cell number determined for each well—and obtained a stimulation factor of ~3-fold (Fig 6B). This result provides a standard for the sorting efficiency and stimulability of a professional granule cargo in AtT20 cells. Similar stimulation experiments performed in PC12 cells (derived from

pheochromocytoma of the rat adrenal medulla), another cell line frequently used to study the regulated pathway, even showed less than twofold stimulation of CgA secretion (Delestre-Delacour et al, 2017).

Stimulated secretion experiments were performed with our AtT20 cell lines expressing the A1Pimyc constructs (Fig 6C and D). Secretion of the reporter A1Pimyc alone was not stimulated by $BaCl_2$ incubation. In contrast, all CC fusion proteins, except CCpN-A1Pimyc, were stimulated significantly by 50–90%, although less than the professional regulated protein CgA. Again, the presence of two copies of a CC loop in the construct did not generally increase the effect. The only exception was CCpN, for which two copies resulted in significant stimulation. With this one exception, the results of stimulated secretion are consistent with the granule localization assay, indicating that increased concentration in the granule tips is due to sorting into functional secretory granules.

## CC loops mediate lubrol insolubility

Aggregation of hormones into secretory granules is transient to allow rapid dissolution upon release into the extracellular medium. Dannies and colleagues (Lee et al, 2001; Zhu et al, 2002) empirically found conditions that solubilized most cellular proteins, but retained the condensed granule cargo largely in an insoluble, pelletable state, using Lubrol as the detergent. As a positive control for Lubrol insolubility experiments, we first tested pro-opiomelanocortin (POMC), an endogenous professional regulated cargo (Fig 7A and B). AtT20 cells were incubated with 1.5% Lubrol for 1 h at 4°C. After a low-speed centrifugation, the post-nuclear supernatant was ultra-centrifuged for 1 h at high speed. Equal fractions of pellet and supernatant were analyzed by immunoblotting for POMC (Fig 7A). Two

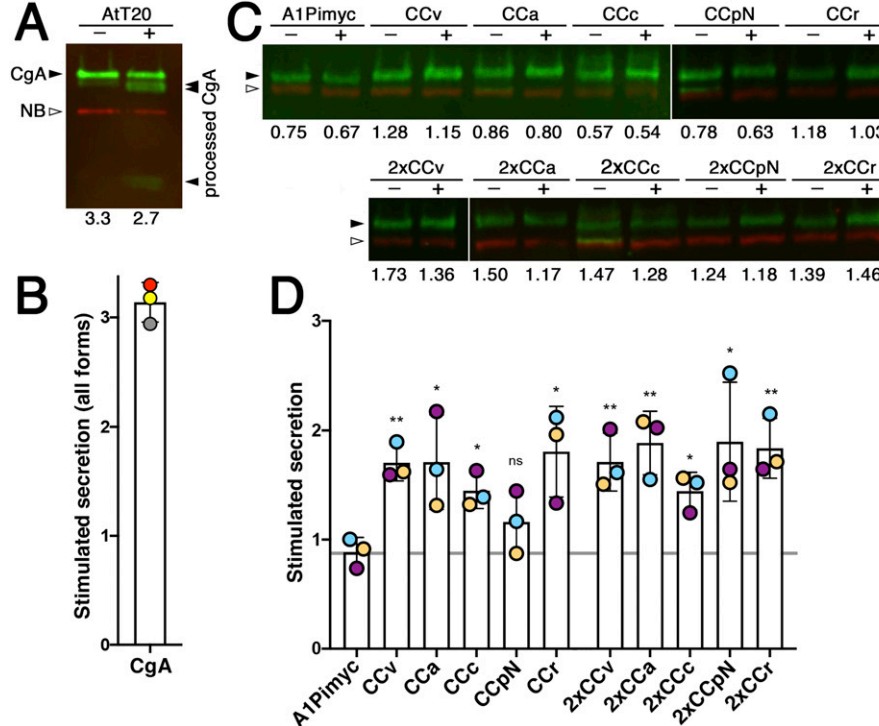

**Figure 6. CC loops mediate stimulated secretion via functional secretory granules.**
**(A)** Parental AtT20 cells were grown in two separate wells, of which one was incubated for 1 h with secretion medium free of divalent cations (–) and the other with stimulation medium containing $BaCl_2$ (+), both containing VHH-mCherry nanobodies (NB) as a loading control. The collected media were subjected to quantitative near-infrared immunoblot analysis for endogenously expressed CgA. The bands in green (filled arrowheads) represent full-length CgA and proteolytically processed forms produced in secretory granules, the red band is the nanobody (open arrowhead). The numbers below the lanes represent the cell count (in $10^6$ per well) for normalization.
**(B)** Stimulated secretion was determined by quantifying the intensities of all CgA forms normalized to the nanobody signal and the cell number, and expressed as the ratio of stimulated to non-stimulated values. Average and SD of three independent experiments are shown and individual values indicated.
**(C)** Stimulated secretion experiments as in panel (A) were performed with the cell lines expressing the indicated A1Pimyc constructs for 30 min. The bands in green and red represent the A1Pimyc constructs and the nanobody, respectively. **(D)** Stimulated secretion was quantified as in (B). Average and SD of three independent experiments are shown, with individual values colored by experiment. Statistical significance was calculated using the unpaired $t$ test versus the A1Pimyc control. ns, nonsignificant; *$P ≤ 0.05$; **$P ≤ 0.01$.

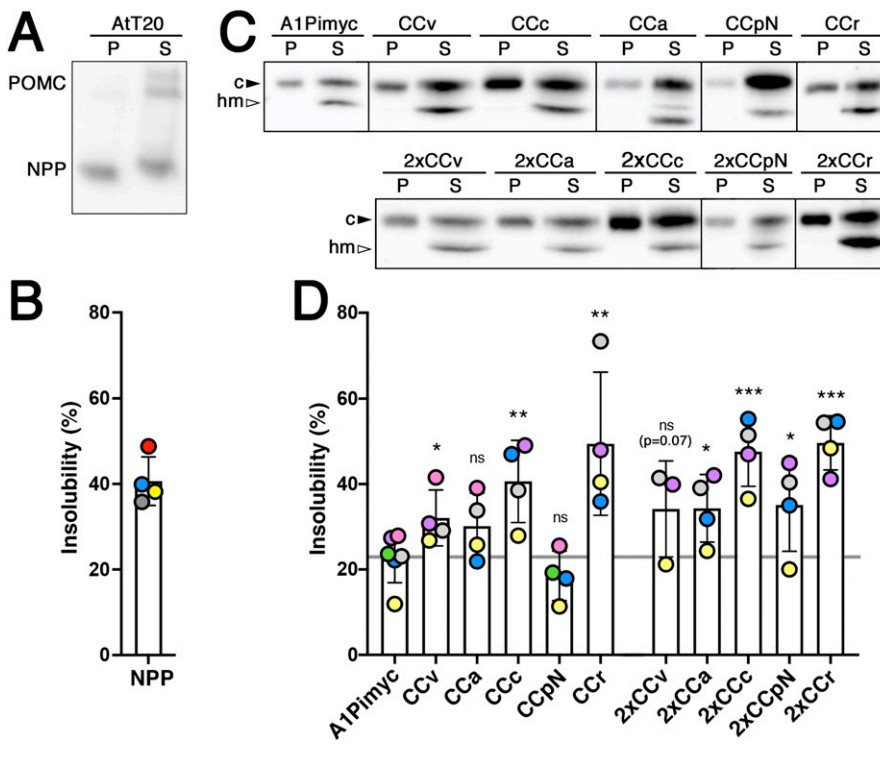

**Figure 7. Insolubility of POMC and CC-A1Pimyc constructs to Lubrol.**
**(A)** AtT20 cells were incubated for 1 h with 1.5% Lubrol at 4°C. The post-nuclear supernatant was centrifuged at 50,000*g* for 1 h. Equal fractions of supernatant (S) and pellet (P) were analyzed by immunoblotting for endogenous POMC. Whereas full-length POMC forms were mostly detected in the supernatant, a large fraction of processed NPP was pelleted. **(B)** The percentage of Lubrol-insoluble NPP was quantified. Average and SD of four independent experiments are shown. **(C)** Lubrol extractions were performed as in (A), using anti-myc antibodies. To better separate high-mannose glycosylated ER (hm) from complex glycosylated Golgi/post-Golgi forms (c) of the A1Pimyc constructs on the gel, the samples were incubated with endoglycosidase H to deglycosylate the high-mannose forms. **(D)** The percentage of Lubrol-insoluble complex glycosylated forms of the proteins was quantified. The average and SD correspond to four independent experiments, except for 2xCCv with three experiments, with individual values colored by experiment. Statistical significance was calculated using the unpaired *t* test. ns, nonsignificant; *$P \leq 0.05$; **$P \leq 0.01$.

high-molecular weight forms of uncleaved POMC and a smaller processed form corresponding to NPP (N-terminal peptide of POMC) were detected. The high-molecular weight forms were recovered mostly in the supernatant, whereas the processed form was to a similar extent found in the insoluble pellet fraction. This selectivity for processed POMC in the pellet thus confirms granule specificity of insolubility. Quantitation indicated ~40% of processed POMC to be insoluble to Lubrol (Fig 7B), which is similar to ~50% insolubility previously reported for growth hormone in $GH_4C_1$ cells (Lee et al, 2001).

We then performed this experiment with all our A1Pimyc constructs (Fig 7C and D). BecauseSince they are N-glycosylated, the EndoH-sensitive high-mannose glycosylated ER form can be distinguished from the EndoH-resistant complex glycosylated form of the Golgi, TGN, and post-Golgi compartments. To better separate the two glycoforms of A1Pimyc and derivatives, the soluble and insoluble fractions were incubated with EndoH for 1 h before immunoblot analysis (Fig 7C). Whereas the ER forms of all constructs were completely solubilized by Lubrol, specifically the complex forms were recovered in the insoluble fraction to variable extents. For A1Pimyc, which is our negative control, ~23% of the complex form was insoluble, likely corresponding to the background of the constitutive secretory protein trapped in granule aggregates. A1Pimyc fused to one or two CC loops increased insolubility of 30–50%, again with the exception of CCpN-A1Pimyc, whereas the increase of CCa-A1Pimyc was not statistically significant (Fig 7D). These Lubrol insolubility experiments thus confirm the localization and stimulation studies to conclude that different CC loops have the ability to reroute A1Pimyc into secretory granules.

### Mutation of the cysteines in a CC loop abolishes granule sorting

To test specificity of CC loop-mediated granule sorting for the disulfide bond, we mutated both cysteines of the generally most active sequence CCc to prolines in construct PPc-A1Pimyc (Fig 8A). A stable AtT20 cell line, expressing PPc-A1Pimyc was isolated with similar expression levels, as determined by metabolic labeling (Fig S2C). The localization of PPc-A1Pimyc was analyzed by immunofluorescence microscopy (Fig 8B). PPc-A1Pimyc showed the same low signal in granule tips as A1Pimyc, as was confirmed by quantitation (Fig 8C), demonstrating that the cysteines and their disulfide bond was required for granule sorting activity.

## Discussion

In the case of provasopressin, granule sorting by self-aggregation of the wild-type protein and ER aggregation of diabetes insipidus mutant proteins were brought together by the amyloid hypothesis of secretory granule biogenesis proposed by Riek and colleagues (Maji et al, 2009; Beuret et al, 2017). It suggested that motifs that evolved for functional aggregation in the TGN may cause pathological aggregation of mutant proteins in the ER. Accordingly, to test a potential general role of CC loops in aggregation, we tested CC loops of different other hormones for their ability to aggregate in both situations.

We analyzed the ER aggregation potential of CC loops upon fusion to a misfolded reporter, NPΔ, using immunofluorescence and immunogold electron microscopy. By light microscopy of transfected COS-1 and Neuro-2a cells, five and three out of six CC

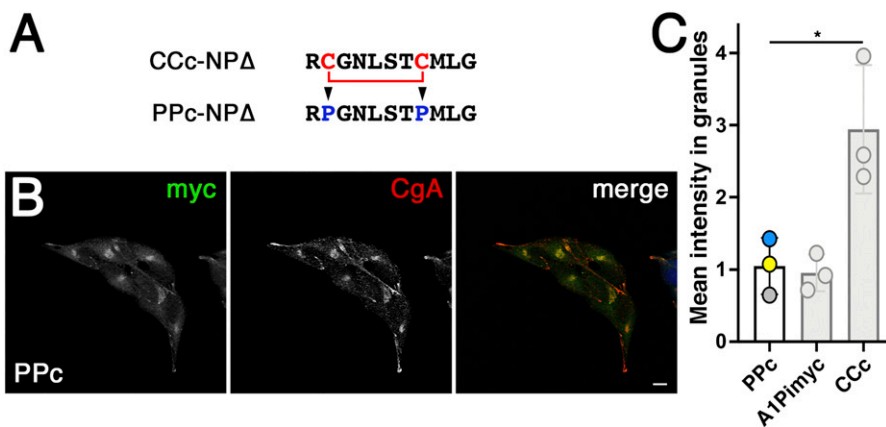

**Figure 8.** **The cysteines are necessary for CCc-A1Pimyc to be sorted into secretory granules.** **(A)** The cysteines of CCc where mutated to prolines in PPc-A1Pimyc. **(B)** PPc-A1Pimyc cells with similar expression levels as A1Pimyc in control cells (shown in Fig S2C) were stained, and imaged as in Fig 5. Scale bar, 10 μm. **(C)** For quantitation, PPc-A1Pimyc expressing cells were grown together with A1Pimyc+EBFP cells on the same coverslip, stained, and imaged, and the myc signal in granules at the cell tips quantified and normalized to that of A1Pimyc in A1Pimyc+EBFP cells on the same coverslip. 20 images with two to eight cells (~100 cells total) of each of them were analyzed per coverslip in three independent experiments. For comparison, the results for A1Pimyc and CCc-A1Pimyc from Fig 5C are shown in gray. Statistical significance was determined using unpaired $t$ test. *$P ≤ 0.05$.

loops, respectively, produced a significant increase of cells with visible accumulations in the ER compared with the control construct, CCv, CCa, and CCpN being the most efficient. However, electron microscopy in Neuro-2a cells detected immunogold decorated dense structures for all CC-NPΔ constructs, although with different sizes and shapes. Although this analysis was not quantitative, the structures were by no means rare also for CCc, CCr, and CCpC, but rather smaller, suggesting that their aggregation propensity was likely underestimated in the fluorescence microscopy assay. In contrast, no similar gold labeled structures were found in cells transfected with the control protein, arguing against residual or background aggregation by the reporter alone.

Ultrastructurally, the aggregates of all constructs appeared equally dense and did not allow demonstration of a fibrillar substructure. This is different from the full-length provasopressin mutant ΔE47 expressed in Neuro-2a cells which showed a loosely packed tangle of fibrils (Birk et al, 2009). However, aggregates of the same protein in COS-1 cells appeared denser and fibrils were hardly detectable (Birk et al, 2009). A truncated and myc-tagged version of provasopressin C61myc (corresponding to 1–72myc, very similar to CCv-NPΔ) produced relatively dense aggregates in which fibrillar substructure was still discernible in HN10 neuroblastoma cells (Beuret et al, 2017). Differences in the reporter and in the expressing cell lines thus appear to affect the packing of the protein together with associated chaperones. Fibrillarity is of course also not detectable in secretory granules, for which an underlying amyloid structure was shown (Maji et al, 2009).

NPΔ may be a rather sensitive reporter for ER aggregation because it contains nine cysteines that can form intermolecular disulfide bonds (Birk et al, 2009; Beuret et al, 2017) likely to stabilize aggregation. It may also account for the strong colocalization of PDI with the aggregates in the colocalization experiments. Overall, our results support the hypothesis of CC loops as general aggregation motifs. By extension, one might speculate that mutation of hormone precursors other than provasopressin might also lead to ER aggregation and dominant disease. Isolated growth hormone deficiency type II is a dominant disease that fits this description (Mullis et al, 2002). It is caused by deletion of exon 3 (residues 32–71) resulting in ER retention and aggregation, as well as retention of co-

expressed wild-type hormone (Missarelli et al, 1997; Lee et al, 2000; Binder et al, 2002). It will thus be interesting to test which sequence(s) cause growth hormone aggregation.

To test the CC loops for their capacity to reroute a reporter into secretory granules, three different methods have been used: (i) localization to CgA-positive secretory granules by immunofluorescence microscopy, (ii) stimulated secretion, and (iii) insolubility to Lubrol.

By immunofluorescence, even the negative control A1Pimyc was detected to some extent in CgA-positive secretory granules. This was not surprising because constitutive proteins are not efficiently excluded from immature secretory granules (Dittie et al, 1996; Kim et al, 2006). Importantly, most CC loop fusion constructs (except CCpN) produced significantly increased accumulation in granules compared with the reporter A1Pimyc alone. Sorting into functional granules was confirmed by stimulated secretion assays. Endogenous regulated cargo that is processed in secretory granules allows distinguishing between the fractions of the proteins that take the regulated versus the constitutive route. The control experiment with endogenous CgA informed on the efficiency of sorting in the AtT20 cells, confirming that a sizable portion of CgA remained uncleaved and was also not stimulated for secretion. Our reporter constructs are not processed. Nevertheless, we observed significant stimulated secretion for all CC loops, except CCpN, in the range of 1.5–2-fold, compared with threefold for CgA when all forms are included. Lubrol insolubility as an empirical assay for granule cargo (Lee et al, 2001; Zhu et al, 2002) further supported enhanced inclusion of CC-A1Pimyc constructs into granules.

Taken together, CC loops were found to be able to direct A1Pimyc, a constitutive reporter, into the regulated pathway and thus to constitute a general type of signal for granule sorting, most likely by aggregation. The different CC loops displayed different efficiencies in the process: CCc and CCr were the most, CCpN the least efficient. This may reflect their inherent activity, but perhaps also differences in accessibility when fused to the globular reporter.

Strikingly, the different CC loop sequences share no sequence similarities. The only obvious common feature is the disulfide bond forming a small ring structure. Experimentally, we found the disulfide bond to be completely formed in the secreted fusion

proteins with CCc, CCa, CCr, and CCv. Upon mutation of the cysteines in CCc to prolines (PPc-A1Pimyc), granule targeting was reduced to the background of the reporter alone. Previously, mutation of cysteines 8 and 20 of POMC also abolished granule sorting (Cool et al, 1995). More experiments have to be performed to identify the important residues within the CC loop sequences.

Of the CC loop-containing hormones, vasopressin and its close relative oxytocin, as well as prolactin and somatostatin-14 have been analyzed by Maji et al (2009) and scored positive at least in several of the amyloid criteria. A hallmark of typical amyloids is their stability and insolubility. Upon stimulated release into the blood stream, hormones must be able to dissolve sufficiently fast to exert their function at the target cells. In vitro aggregates were shown to be able to redissolve within several hours (Maji et al, 2009), demonstrating that they are much less insoluble than the typical pathogenic amyloids. The small disulfide loops we analyzed here cannot form an amyloid in the strict sense. They likely aggregate in an "amyloid-like" manner with backbone–backbone hydrogen bonds and sidechain–sidechain packing of fibrous elements to account for the amyloid diffractions. CC loop aggregates are likely to be less stable and thus more soluble under appropriate conditions. Indeed, in vitro aggregation studies with somatostatin-14, the C-terminal 14 residues of prosomatostatin with a Cys-3–Cys-14 disulfide loop, showed that reduction of the disulfide bond increased aggregation rate and reduced re-dissolution kinetics (Anoop et al, 2014). This may suggest that CC loops produce "poor" amyloids by design, sufficient to aggregate in the TGN, but also to be able to re-dissolve quickly upon secretion.

Granule formation at the TGN involves aggregation of natively folded (precursor) proteins. For amyloid or amyloid-like fibers to form, a sufficiently long peptide segment needs to be exposed or, alternatively, the protein must first unfold and then refold upon secretion. In the case of provasopressin with two aggregating sequences, the glycopeptide appears to be exposed, whereas vasopressin folds into a binding pocket in NPII (Chen et al, 1991; Wu et al, 2001). Changing conditions in the TGN may trigger release from its binding site without unfolding.

Similarly, we could postulate that CC loops should generally have a position in the protein that allows them to be easily exposed. Based on the structure, the N-terminal CC loop of prolactin is naturally exposed and the C-terminal one, which is structurally shared with growth hormone, also appears to be exposable with minimal conformational changes. In contrast, the CC loop of pro-renin is neither at the N- nor C-terminal end but at the extremity of one of its lobes, suggesting that a small conformational change might expose it sufficiently as well.

Not all prohormones contain CC loops, which are thus not the only and universal granule signal. Interestingly, CgB contains a larger disulfide loop of 22 amino acids that has previously been shown to be sufficient for sorting into secretory granules (Krömer et al, 1998; Glombik et al, 1999) and, similarly, the homologous N-terminal domain of CgA {Taupenot:2002p2031}. Furthermore, the same CgA domain induced granule-like structures in COS-1 cells (Stettler et al, 2009). This suggests that not only short disulfide loops, but also longer ones may play a role in granule sorting.

Taken together, our data support the hypothesis that CC loops of different peptide hormone precursors have the ability to cause aggregation and to reroute a constitutive reporter into secretory granules. They thus act as novel signals for granule sorting.

# Materials and Methods

## Antibodies

As primary antibodies, we used mouse monoclonal anti-myc (precipitated from 9E10 hybridoma cultures, RRID:CVCL_L708, 1:10 to 1:100 for immunofluorescence), rabbit monoclonal anti-myc (Genetex, GTX29106, Lot:821902572, RRID:AB_369669, 1:400 for immunofluorescence), mouse monoclonal anti-His$_6$ (HIS.H8, Millipore, MAI-21315, Lot: 2426469, RRID:AB 557403, 1:2000 for immunofluorescence), rabbit polyclonal anti-CgA (Novus Biologicals, NB120-15160, Lot: C-2, RRID:AB_789299, 1:200 for immunofluorescence and 1:5,000 for immunoblot), monoclonal mouse anti-actin (clone C4, Millipore, MAB1501, Lot:3282535, RRID:AB_2223041, for immunoprecipitation), rabbit monoclonal anti-His$_6$ (Abcam, Ab213204, Lot:GR3199118-5, for electron microscopy), rabbit monoclonal anti-POMC (Abcam, ab254257, GR3279295-2, 1:5,000 for immunoblot); mouse monoclonal anti-HA (precipitated from 12CA5 hybridoma cultures, 1:2,000 for Western blot), and rabbit monoclonal anti-HA antibodies (Cell Signaling Technology, C29F4, Lot: 9, RRID:AB_1549585, 1:5,000 for immunoblot).

As secondary antibodies, we used non-crossreacting A488-labeled donkey anti-mouse immunoglobulin (Molecular Probes, A21202, Lot: 1975519, RRID: AB_141607, 1:300 for immunofluorescence), A488-labeled donkey anti-rabbit immunoglobulin (Molecular Probes, A21206, Lot: 1981155, RRID: AB_2535792, 1:300 for immunofluorescence), HRP-labeled goat anti-mouse Fc (Sigma-Aldrich, A0168, Lot:024M4751. RRID: AB_257867, 1:20,000 for immunoblot), HRP-labeled goat anti-rabbit immunoglobulin (Sigma-Aldrich, A0545, Lot:022M4811, RRID: ab_257896, 1:20,000 for immunoblot), IR Dye800CW-labeled donkey anti-rabbit immunoglobulin (Li-Cor, 926-32213, Lot:C70405-07, RRID: AB_621848, 1:20,000 for near-infrared immunoblot), IDRye680RD-labeled donkey anti-mouse immunoglobulin (Li-Cor, 926-68072, Lot:C80522-25, RRID: AB_10953628, 1:20,000 for near-infrared immunoblot), and goat anti-rabbit immunoglobulin antibodies (BB International, EM GAR10, RRID: AB_1769128, 1:100 for electron microscopy).

## Plasmids and constructs

All constructs were cloned into the mammalian expression vector pcDNA3 (Invitrogen) with geneticin (A1Pi constructs) or a pcDNA3 derivative with the selection gene replaced by that for hygromycin (EBFP). The CCv-NPΔ and Pro1-NPΔ were described by Beuret et al (2017) as 1–75 and 1–75Pro1, respectively. The other CC-NPΔ constructs were produced as complementary synthetic oligonucleotides and inserted between the coding sequences of the signal peptide and neurophysin II. The same CC sequences were inserted between the signal peptide of prepro-enkephalin and the mature sequence of A1Pi variant M2 with a C-terminal six-histidine (His$_6$) and myc epitope tag for CC-A1Pimyc, and additionally also between the C terminus of A1Pi and the His$_6$ and myc tags for 2xCC-A1Pimyc. The cDNA of human PDI (from Dr. Julia Birk, University of Basel) was

tagged at the N terminus with a myc epitope by polymerase chain reaction.

VHH-mCherry nanobodies (anti-GFP nanobodies fused to mCherry and T7, HA and hexahistidine tags) were kindly provided by Dr. Dominik Buser (Addgene plasmid # 109421; http://n2t.net/addgene:109421; RRID: Addgene_109421) (Buser et al, 2018).

## Cell culture and transfection

Mouse neuroblastoma cells Neuro-2a (kind gift by Dr. Ling Qi, University of Michigan), monkey kidney fibroblast-like COS-1 cells (from Dr. Richard E. Mains, University of Connecticut) and mouse pituitary corticotrope tumor cells AtT20 (from Dr. Hans-Peter Hauri, Biozentrum) were grown in DMEM (Sigma-Aldrich) containing 4,500 mg/l (Neuro-2a) or 1,000 mg/l glucose (AtT20 and COS-1), supplemented with 10% FCS, 100 units/ml penicillin, 100 $\mu$g/ml streptomycin, and 2 mM L-glutamine at 37°C with 5% (Neuro-2a) or 7.5% $CO_2$ (AtT20 and COS-1). Cells were transfected using Fugene HD (Promega).

Stable AtT20 cell lines expressing A1Pi constructs were selected using 100 $\mu$g/ml G418-sulfate (Invivogen). EBFP cloned into pcDNA3 (hygromycin B as selective antibiotic) plasmid was transfected into A1Pimyc expressing cell line, selected with 180 $\mu$g/ml hygromycin B (Invivogen). Stable clonal cell lines were isolated by dilution into 96-well plates from the pooled stable cell lines.

## Immunofluorescence microscopy and quantitation

Cells were grown on glass coverslips to 50–60% confluence, fixed with 3% paraformaldehyde for 30 min at room temperature, washed twice with PBS, quenched 5 min in 50 mM $NH_4Cl$ in PBS, washed twice with PBS, permeabilized in 0.1% Triton X-100 (Applichem) in PBS for 10 min, washed twice with PBS, blocked with 1% BSA (Roche) in PBS for 15 min, incubated at room temperature with primary antibodies for 1–2 h in BSA-PBS, washed twice with PBS, stained with fluorescent secondary antibodies in BSA/PBS for 30 min, and mounted in Fluoromount-G (Hoechst) with or without 0.5 $\mu$g/ml DAPI. The stainings were analyzed using a Zeiss Confocal LSM700 microscope.

For quantitation of cells with ER aggregates, Neuro-2a cells and COS-1 cells transfected with Pro1-NPΔ or CC-NPΔ constructs were analyzed by fluorescence microscopy using a Zeiss Axioplan microscope to count the percentage of expressing cells showing punctate accumulations. Between 120 and 150 expressing cells were counted for each construct in each of three independents experiments.

For quantitation of myc-tagged protein in secretory granules, AtT20 cell lines expressing A1Pimyc, CC-A1Pimyc, and 2xCC-A1Pimyc were co-seeded with A1Pimyc+EBFP cells on coverslips and subjected to immunofluorescence analysis using mouse anti-myc and rabbit anti-CgA antibodies. Intensity in granules was quantified using Fiji and a script by Dr. Laurent Gerard (Biozentrum). CgA staining was used to manually identify the secretory granules in the tips of the cells, where the script determined the mean intensity of myc signal in CgA-positive pixels. Per coverslip, ~10 images were taken each for A1Pimyc+EBFP and for the construct of interest with between 5 and 8 cells per picture, corresponding to 80–150 tips quantified per construct. The mean intensity of anti-myc signal in granule tips was normalized to the number of tips and to the normalized intensity of A1Pimyc in the A1Pimyc+EBFP cells on the same coverslip. Three independents' experiments were performed for each cell line.

## Immunogold electron microscopy

Transfected Neuro-2a cells were grown in 10-cm dishes, transiently transfected with the NPΔ constructs and grown until 80% confluence. Cells were fixed in 3% formaldehyde and 0.2% glutaraldehyde for 2 h at room temperature, then scraped, pelleted, resuspended, and washed three times in PBS, incubated with 50 mM $NH_4Cl$ in PBS for 30 min, washed three times in PBS, resuspended in 2% warm agarose, and left to solidify on ice. Agarose pieces were dehydrated, infiltrated with LR gold resin (London Resin), and allowed to polymerize for 1 d at −10°C. Sections of 60–70 nm were collected on carbon-coated Formvar Ni-grids, incubated with rabbit anti-His$_6$ antibodies in PBS, 2% BSA, 0.1% Tween-20 for 2 h, washed with PBS, and incubated with 10-nm colloidal gold-conjugated goat anti-rabbit immunoglobulin antibodies in PBS, 2% BSA, and 0.1% Tween-20 for 90 min. Grids were washed five times for 5 min in PBS and then five times in $H_2O$, before staining for 10 min in 2% uranyl acetate. Sections were imaged with a Phillips CM100 electron microscope.

## Radioactive labeling

To determine the expression levels, AtT20 cells stably expressing A1Pimyc constructs were grown in six-well plates to 80% confluence, incubated at 37°C for 30 min in 500 $\mu$l starvation medium (DMEM lacking methionine and cysteine) and another 30 min in 500 $\mu$l labeling medium containing 100 $\mu$Ci/ml [$^{35}$S]methionine/cysteine (Perkin Elmer). The cells were washed twice with PBS and lysed in 500 $\mu$l lysis buffer (PBS with 1% Triton X-100, 0.5% deoxycholate, and 2 mM phenylmethylsulfonyl fluoride, pH 8) for 1 h at 4°C. The lysates were scraped and centrifuged for 20 min in a microfuge at 13,000$g$. 400 $\mu$l of supernatants were incubated overnight at 4°C with 20 $\mu$l of mouse monoclonal anti-myc antibodies. Upon addition of 20 $\mu$l protein A Sepharose beads (Bio-Vision), samples were incubated on a shaker for 1 h at 4°C, washed three times with lysis buffer, once with 100 mM Na-phosphate, pH 8, and once with 10 mM Na-phosphate, pH 8. Upon addition of 40 $\mu$l 2xSDS-sample buffer and 1/8 volume of 1 M DTT, samples were heated at 95°C for 5 min and analyzed by SDS-polyacrylamide gel electrophoresis (10% acrylamide). Gels were fixed, dried, exposed to phosphorimager plates for 2–5 d, and analyzed by phosphorimager.

## Stimulated secretion

Parental AtT20 cells or AtT20 cells stably expressing A1Pimyc constructs were grown in 12-well plates until 80% confluence. Cells were washed with PBS, incubated for 30 min or 1 h either with 150 $\mu$l of resting secretion medium (Earle's balanced salt solution without $Ca^{2+}$ and $Mg^{2+}$, E6267, Sigma-Aldrich, supplemented with MEM amino acids, 2 mM L-glutamine, 97.65 mg/l $MgSO_4$, containing 2 $\mu$g/ml VHH-mCherry nanobodies for normalization) or with 150 $\mu$l of stimulation medium (secretion medium supplemented with 2 mM

barium chloride, BaCl$_2$). Both media were collected, separated by SDS-gel electrophoresis, and subjected to quantitative near-infrared immunoblot analysis.

### Lubrol solubility assay

AtT20 cell lines stably expressing A1Pimyc constructs were grown in six-well plates to 80% confluence, washed 3 times with cold PBS, covered with 200 µl 1.5% Lubrol (MP Biomedicals) in 5 mM Na-phosphate, pH 7.4, 0.3 M sucrose, 2 mM PMSF, and protease inhibitor cocktail, scraped into a centrifuge tube, incubated for 1 h at 4°C on a roller, and then centrifuged for 5 min at 500g at 4°C. 150 µl of the supernatant was centrifuged for 1 h at 50,000g at 4°C. The supernatant was collected and supplemented with 37.5 µl 5xSDS-sample buffer, whereas the pellet was resuspended in 187.5 µl 1xSDS sample buffer. Both fractions were heated for 3 min at 95°C. To analyze A1Pimyc constructs, 500 units endoglycosidase H (New-England Biolabs) was added to 40-µl aliquots of supernatant and pellet and incubated 1 h at 37°C. After addition of 1/8 volume of 1 M DTT, samples were boiled for 2 min at 95°C, separated on SDS-gels containing 10% or 7.5% acrylamide for A1Pi constructs or POMC, respectively, and analyzed by immunoblotting.

### Mass spectrometry

AtT20 cell lines stably expressing A1Pimyc, CCv-, CCc-, CCa-, CCr-, or CCpN-A1Pimyc were grown in 10-cm dishes, washed twice with PBS, and incubated overnight with 4 ml medium without FCS. The medium was centrifuged for 1 min at 13,000g and the supernatants incubated overnight with 40 µl of anti-myc antibodies and incubated with 25 µl protein A Sepharose beads (BioVision) for 1 h at 4°C on a shaker. The beads were washed four times with 100 mM and then 10 mM Na-phosphate, and the antigen was eluted with 80 µl of 0.2 M glycine, pH 2.8. Samples were split in two tubes of 40 µl each. After neutralization with 20 µl ammonium bicarbonate, one tube of each sample was reduced by addition of 5 µl tris(2-carboxyethyl) phosphine (TCEP). To both tubes, 5 µl digestion buffer (preparation for 4 ml: 2.3 g guanidine-HCl, 400 µl 1 M ammonium bicarbonate, 1 ml 0.75 M chloroacetamide) was added to alkylate free thiol groups. Samples were heated at 95°C for 10 min and then supplemented with either 0.5 µg trypsin (A1Pimyc, CCv-A1Pimyc, CCa-A1Pimyc, and CCc-A1Pimyc) or 0.5 µg Lys-C (CCr-A1Pimyc and CCpN-A1Pimyc) incubated overnight at 37°C. 50 µl 2 M hydrochloric acid and 50 µl 5% trifluoroacetic acid were added and peptides desalted using BioPureSPN Mini C18 spin columns (Part #HUMS18V from The Nest Group). After drying, 0.2 µg of peptides were LC–MS analyzed as recently described (Ahrné et al, 2016) using a Q-Exactive Plus Orbitrap mass spectrometer with a nanoelectrospray ion source (Thermo Fisher Scientific). Each MS1 scan was followed by high-collision-dissociation of the precursor masses of the 20 most abundant precursor ions with dynamic exclusion for 20 s. Total cycle time was ~1 s. For MS1, $3 \times 10^6$ ions were accumulated in the Orbitrap cell over a maximum time of 50 ms and scanned at a resolution of 70,000 FWHM (at 200 m/z). MS1-triggered MS2 scans were acquired at a target setting of $1 \times 10^5$ ions, a resolution of 17,500 FWHM (at 200 m/z) and a mass isolation window of 1.4 Th. Singly charged ions and ions with unassigned charge state were excluded

from triggering MS2 events. The normalized collision energy was set to 27% and one microscan was acquired for each spectrum. The spectral library was exported into Skyline (v21.1.0.278) (https://brendanx-uw1.gs.washington.edu/labkey/project/home/software/Skyline/begin.view). Precursor ions were selected as transition filters and all raw-files imported for quantification. The results of the peptide ions containing the CC-loop peptides covering the two cysteines in their disulfide bonded or twice carbamidomethylated (alkylated) forms. Unfortunately, neither of the two forms of the peptide expected for CCpN-A1Pimyc could be detected. All calculations were performed in Excel.

## Supplementary Information

## Acknowledgements

We thank Drs. Laurent Guerard and the Biozentrum Imaging Core Facility, and Alexander Schmidt of the Proteomics Core Facility for their support. This work was supported by Grant 31003A-182519 from the Swiss National Science Foundation.

### Author Contributions

J Reck: conceptualization, validation, investigation, visualization, and writing—original draft, review, and editing.
N Beuret: conceptualization and investigation.
E Demirci: investigation.
C Prescianotto-Baschong: investigation.
M Spiess: conceptualization, supervision, funding acquisition, visualization, project administration, and writing—original draft, review, and editing.

### Conflict of Interest Statement

The authors declare that they have no conflict of interest.

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
