## [Reviewer comments · Life Science Alliance]

Life Science Alliance

Small disulfide loops in peptide hormones mediate self-aggregation and secretory granule sorting

Jennifer Reck, Nicole Beuret, Erhan Demirci, Cristina Prescianotto-Baschong, and Martin Spiess

DOI: <https://doi.org/10.26508/lsa.202101279>

Corresponding author(s): *Martin Spiess, University of Basel*

Review Timeline:

Submission Date:	2021-10-27
Editorial Decision:	2021-10-27
Revision Received:	2021-12-13
Editorial Decision:	2022-01-11
Revision Received:	2022-01-12
Accepted:	2022-01-14

Scientific Editor: *Eric Sawey, PhD*

Transaction Report:

Please note that the manuscript was reviewed at *Review Commons* and these reports were taken into account in the decision-making process at *Life Science Alliance*.

October 27, 2021

Re: Life Science Alliance manuscript #LSA-2021-01279

Prof. Martin Spiess
University of Basel
Biozentrum
Spitalstrasse 41
Basel, CH CH-4056
Switzerland

Dear Dr. Spiess,

Thank you for submitting your manuscript entitled "Small disulfide loops in peptide hormones mediate self-aggregation and secretory granule sorting" to Life Science Alliance. We invite you to re-submit the manuscript, revised according to your Revision Plan.

When submitting the revision, please include a letter addressing the Reviewers' comments point by point.

Thank you for this interesting contribution to Life Science Alliance. We are looking forward to receiving your revised manuscript.

Sincerely,

B. MANUSCRIPT ORGANIZATION AND FORMATTING:

Life Science Alliance manuscript #LSA-2021-01279**Point-by-point response**

Reviewer #1 (Evidence, reproducibility and clarity (Required)):

Apart from the default constitutive pathway for protein secretion some specialized cells (e.g., neuroendocrine cells, exocrine cells, peptidergic neurons and mast cells) exhibit additional regulated secretory pathway, where peptide hormones are stored as highly concentrated ordered manner inside electron opaque "dense core" of secretory granule for long duration until secretagogue mediated burst release. Although the general sorting receptor for packaging hormones in secretory granules is not yet identified, self-aggregation in the trans-Golgi network is a common shared property of peptide hormones and is a well-accepted potential sorting mechanism. Here the authors have hypothesized that cysteine containing small disulphide loop (CC loop), which is abundant in several hormone precursors, acts as aggregation mediator in TGN for sorting into secretory granule. They have tested the aggregation propensity of a misfolded reporter protein, NPΔ, in ER by attaching the CC loop segment of different hormones which promoted the pathological aggregation in endoplasmic reticulum (ER) of mutant provasopressin in the case of diabetes insipidus. Immunofluorescence and immunogold electron microscopy revealed accumulation of aggregates in the ER when CC loop of different hormonal origin fused NPΔ was transiently transfected in COS-1 fibroblasts and Neuro-2a neuroblastoma cells. The authors have also shown small disulphide loop mediated functional aggregation in TGN can sort a constitutively secreted protein, α1-protease inhibitor, into the secretory granule. The rerouting capacity of CC loop was tested in stably expressed AtT-20 cell line by confirming their localization with CgA-positive secretory granule as well as by studying BaCl₂ mediated stimulated secretion and by testing secretory granule specific lubrol insolubility.

****Major comments:****

The study is highly impressive, and the results fully support the CC loop mediated hormone sorting hypothesis. However, it would be nice if the authors characterize the nature of the CC-loop mediated aggregates as hormones are reported to be stored inside secretory granules as functional amyloid (Maji et al., 2009). The mechanistic reason behind the small disulphide loop mediated aggregation was not explained in the paper. Authors may propose the probable molecular reasons behind CC loop mediated aggregation to completely justify their hypothesis.

The starting point for our analysis was the study by Maji et al. (2009) showing that granule cargo including the CC loop-containing vasopressin, oxytocin, somatostatin, and prolactin form amyloids in vitro and that granules are amyloids. Our own studies Birk et al. (2009) and Beuret et al. (2017) found fibrillar aggregation of mutant provasopressin in vitro and CCv-NPΔmyc in the ER of expressing cells. To go further, we focused in the present study on the ability of CC loops in general to aggregate in the ER and to sort by aggregation into secretory granules in vivo, rather than on the amyloid(-like) character again.

Although the hypothesis and the experimental results are highly impressive, the authors may consider adding the following experiments.

The authors replaced CC-loop by the proline/glycine repeat sequence (Pro1) as a negative control which was previously reported to abolish aggregation as well. However, the authors may completely delete the small loop forming segment, CCv, and may check the status of His-tagged fused neurophysin II (NPΔ) segment as an additional negative control.

As suggested, we used the NPΔ construct completely lacking any N-terminal extension as a further negative control. As expected, very similar levels of aggregation were measured as for Pro1-NPΔ. We have included this additional control construct in Figure 1 and 2 and we changed the text accordingly.

To find the ultrastructure authors have done immunogold assay with anti-His antibody which indicated different CC loop mediated ER aggregation. Since the amyloid-like fibril nature of pro-vasopressin mutant mediated ER aggregates was previously reported (Beuret et al., 2017), authors must check the nature of the CC loop mediated ER aggregates with amyloid specific antibody.

We have previously tested staining of ER aggregates of diabetes insipidus mutants of provasopressin with the anti-amyloid antibodies and Thioflavin – without success. Motivated by the reviewers' proposal, we again attempted staining with anti-amyloid OC antibodies and with Thioflavin, this time on CC-NPΔ ER aggregates. No signal was obtained with either method. Our attempts to stain ER aggregates might have been unsuccessful because the CC loops do not produce typical amyloids, since the ring closure prevents a perfect β conformation, or because disulfide crosslinking of the unfolded NPΔ reporter (Birk et al., 2009) hinders antibody access.

Since hormones are known to form reversible functional amyloid during their storage inside secretory granule, authors may consider characterizing the nature of the aggregates formed by CC loop fused constitutive protein in AtT-20 cell line by

immunostaining, immunoprecipitation and dot blot assay using amyloid specific antibody.

Like for ER aggregates, using anti-amyloid OC antibodies or Thioflavin, no signal could be detected in endogenous secretory granules of Neuro-2a and AtT20 cells with or without expression of CC-A1Pimyc. Maji et al. (2009) is the only study we know where Thioflavin stained granules in fluorescence microscopy. Here, sections of pituitary tissue were analyzed. It seems likely that the density and amounts of granules are much higher than in cultured AtT20 or Neuro-2a cells. We are not aware of studies where staining of granules by anti-amyloid antibodies in fixed cells was shown by immunofluorescence microscopy.

****Minor comments:****

In the quantification study (Figure 2C) CCc and CCr showed almost similar ER aggregates (around 40%). But authors have commented that all constructs except CCc produce statistically significant increases in cells compared to background. Authors must clarify the statement.

CCc also increased, but in a statistically not significant manner ($p = 0.08$). We have changed the sentence to: "It confirmed the ability of all constructs to produce an increase of cells with aggregates above background in COS-1 cells (Figure 2C), although not statistically significant for CCc ($p = 0.08$)."

In lubrol insolubility assay, the otherwise constitutively secreted protein A1Pimyc (negative control) showed 23% insolubility. The authors explained the observation by commenting about trapping of the protein inside granule aggregate. But CCv and CCa fused proteins showed a very slight increase (around 30%). Only CCc construct showed more than 40% insolubility. If the trapping of constitutive protein may result in 23% insolubility, all the insolubility data except CCc is not satisfactory to claim as secretory granular content of aggregated protein. The authors must explain that.

Lubrol insolubility is an empirical assay with high specificity for Golgi/post-Golgi forms, but with a relatively high background that we suggest to be due to trapping. Interpretation is based on statistical analysis of several independent experiments. It supports the conclusion of the other assays from an independent angle.

The authors have satisfactorily referenced prior studies in the field. However, authors may consider adding the following papers as they are directly connected with the hypothesis. The sorting of POMC hormone into secretory granules by disulphide loop was previously studied. (Cool et al., 1995). The N-terminal loop segment was also previously used to reroute a constitutive protein chloramphenicol acetyltransferase (Tam and Peng, 1993). S K. Maji and his coworker had previously shown that disulphide bond maintains native reversible functional amyloid structure relevant to hormone storage inside secretory granule whereas disulphide bond disruption led to rapid irreversible amyloid aggregation using cyclic somatostatin as model peptide. (Anoop et al., 2014).

Thank you for pointing out POMC, where the 13-amino acid CC loop is within a 23-amino acid disulfide loop. The recent AlphaFold structure prediction shows the inner loop to be exposed. We have now included POMC in Table I and refer to the supporting studies by Cool et al. (1995) and Tam et al. (1993) in the Introduction.

The study by Anoop et al., 2014, is already discussed in the Discussion.

Authors must check grammar and may reconstruct a few sentences where sentence construction seems complicated.

We went through the text and changed some complicated sentences to improve readability.

Reviewer #1 (Significance (Required)):

This manuscript has a significant contribution to enrich academia with fundamental research knowledge of hormone sorting mechanisms. Although constitutive and regulated secretory pathways are known for long times, the exact sorting mechanism is not yet elucidated. There is no common receptor identified yet for recruiting regulated secretory proteins inside the secretory granules.

Aggregation in the TGN is a well-accepted mechanism for sorting. However, the triggering factor for aggregation is not yet known. This study has shed light on a novel hypothesis, which has considered intramolecular disulfide bond mediated small CC loop in hormone may act as aggregation mediator. Since many regulated secretory proteins contain the short disulphide loop, the hypothesis proposed in the manuscript is interesting.

It has been confirmed that TGN is the last compartment which is common to both regulated and constitutive pathways (Kelly, 1985). There is no sorting mechanism required for the constitutive one as this is the default mechanism, whereas a regulated secretory pathway requires a specific sorting mechanism to be efficiently packaged in the secretory granules. There are two popular hypotheses about protein sorting in regulated secretory pathways. They are "sorting for entry" and "sorting for retention" (Blázquez and Kathleen, 2000). In "sorting for entry" hormones destined to go to the regulated secretory pathway start to form aggregates in the TGN specific environment excluding other proteins destined to go to the constitutive pathway. Arvan and Castle proposed the second mechanism as some hormones, like proinsulin, are initially packaged with lysosomal enzymes in immature secretory granules (ISG) (Arvan and Castle, 1998). But with time they start to aggregate and lysosomal

enzymes are removed from ISG by small constitutive-like vesicles. Although, in both the mechanisms aggregation is an essential sorting criterion the molecular events that lead to aggregation is not yet elucidated. TGN specific environmental conditions including pH (around 6.5), divalent metal ions (Zn²⁺, Cu²⁺), Glycosaminoglycans (GAGs) have potential to trigger aggregation (Dannies, Priscilla S, 2012). Though each hormone has aggregation prone regions in the amino acid sequence, there is no common amino acid sequence responsible for aggregation. The authors in this manuscript, have pointed out an interesting observation that many hormones contain small disulfide loops which are exposed due to their presence in N or C terminal or close to the processing site. Based on their observation, they hypothesized CC loop may act as aggregation driver for hormone sorting. In-cell study with CC construct from different hormones successfully rerouted a constitutively secretory protein into the regulated pathway which supported their novel hypothesis.

However, the hypothesis raises some questions to be answered regarding the molecular mechanism of CC loop mediated aggregation. Why does CC-loop promote aggregation? Does the amino acid sequence, size of the loop play a role in aggregation? The granular structure shown in the manuscript from different CC loops has different size and shape (Figure 2 and 3). What is the reason for the structural heterogeneity of the CC loop mediated dense core? Since authors have shown CC loop mediated aggregation both in functional as well as in diseased aggregation, a very important aspect to address would be the structure-function relationship of the aggregates. Since authors have rightly pointed out that not all hormones or prohormones contain CC loop, another curious question would be about the sorting mechanism of those without CC loop. The best part of the study is that it has tried to explain the well-established aggregation mediated sorting mechanism from a new perspective, which raises room for many questions to be addressed by further research.

From this study, the audience will get to know about the role of small disulphide loop in functional and diseased associated protein/peptide aggregation. The audience will also get an idea about the sorting mechanism in the regulated secretory pathway from the study. According to my expertise and knowledge where I do protein aggregation related to human diseases and hormone storage, I see this manuscript is a fantastic addition to understand the secretory granules biogenesis of hormones with storage and subsequent release.

Reference:

- Maji, Samir K., et al. "Functional amyloids as natural storage of peptide hormones in pituitary secretory granules." *Science* 325.5938 (2009): 328-332.
- Beuret, Nicole, et al. "Amyloid-like aggregation of provasopressin in diabetes insipidus and secretory granule sorting." *BMC biology* 15.1 (2017): 1-14.
- Cool, David R., et al. "Identification of the sorting signal motif within pro-opiomelanocortin for the regulated secretory pathway." *Journal of Biological Chemistry* 270.15 (1995): 8723-8729.
- Tam, W. W., K. I. Andreasson, and Y. Peng Loh. "The amino-terminal sequence of pro-opiomelanocortin directs intracellular targeting to the regulated secretory pathway." *European journal of cell biology* 62.2 (1993): 294-306.

- Anoop, Arunagiri, et al. "Elucidating the Role of Disulfide Bond on Amyloid Formation and Fibril Reversibility of Somatostatin-14: RELEVANCE TO ITS STORAGE AND SECRETION." *Journal of Biological Chemistry* 289.24 (2014): 16884-16903.
- Kelly, Regis B. "Pathways of protein secretion in eukaryotes." *Science* 230.4721 (1985): 25-32.
- Blázquez, Mercedes, and Kathleen I. Shennan. "Basic mechanisms of secretion: sorting into the regulated secretory pathway." *Biochemistry and Cell Biology* 78.3 (2000): 181-191.
- Arvan, Peter, and David Castle. "Sorting and storage during secretory granule biogenesis: looking backward and looking forward." *Biochemical Journal* 332.3 (1998): 593-610.
- Dannies, Priscilla S. "Prolactin and growth hormone aggregates in secretory granules: the need to understand the structure of the aggregate." *Endocrine reviews* 33.2 (2012): 254-270.

Reviewer #2 (Evidence, reproducibility and clarity (Required)):

Summary:

This manuscript by Reck and colleagues aim at determining the importance of short disulfide loops for the correct sorting to, and release from, secretory granules. They utilize hybrid secretory proteins where sequences encoding disulfide loop from different hormones are cloned in frame with the same secretory peptide, and assess how the presence of the disulfide loop affect the ability of the protein to aggregate in the ER and to get sorted for secretion. By immunofluorescence analysis they show that the presence of a disulfide loop increases the ability of the peptide hormone to form aggregates in the ER, and these observations are confirmed by immunogold-EM. Importantly, aggregate formation is seen both in professional secretory (N-2a) and non-secretory (COS-1) cells. Using immunofluorescence and quantitative immunoblotting, they also show that the ability to aggregate the secretory proteins coincide with increased localization to secretory granules and in increased release from cells in response to stimuli.

The results from this study are interesting and suggest that small disulfide loops may be an important part of the cargo sorting mechanism in secretory cells, and perhaps also a cause of sorting defects in certain diseases. The study is overall well conducted and worthy of publication after revision.

****Major comments:****

1) It is unclear to me what the relationship between the CC-loop and amyloid is. They are not involved in the formation of fibrils and amyloid, yet the authors conclude that they support the amyloid hypothesis of granule biogenesis. This must be clarified.

In this study, we do not directly provide evidence for the amyloid (or rather amyloid-like) character of aggregation and therefore the last sentence of the Discussion is not strictly correct. Our findings are consistent with the amyloid hypothesis, but they do not provide additional evidence for the amyloid nature of granules. We have now deleted the last statement of the discussion.

Maji et al. (2009) and Anoop et al. (2014) already showed that the CC loop hormones vasopressin, oxytocin, and somatostatin-14 form fibrillar aggregates in vitro, and that granules have amyloid characteristics. Our own lab showed that folding-deficient mutant forms of vasopressin formed fibrillar aggregates in vitro (Birk et al., 2009) and in the ER of expressing cells (Birk et al., 2009; Beuret et al., 2011) and that vasopressin (CCv) is responsible for this and for granule sorting (Beuret et al., 2011). In our current study unfortunately, the ER aggregates of CC-NPΔmyc (including CCv) in Neuro-2a cells were too compact to reveal fibrillarity. However, secretory granules, which contain functional amyloids, similarly do not have a fibrillar appearance. While in this study we do not provide direct evidence that CC loops produce amyloids, evidence has previously been published.

2) What is the actual function of the CC-loops? The authors show that the loops promote aggregation of cargo proteins, yet the mechanism behind this is unclear. For example, would the proteins used in this study be able to aggregate in vitro (i.e. the CC-loop enable aggregation) or do they require some co-factor/chaperone? It would also be good if the authors could clarify or explain why some CC-loops cause aggregation and others not.

Maji et al. (2009) showed for 3 different CC loops (vasopressin, oxytocin and somatostatin-14) that they aggregate in an amyloid-like manner in vitro in purified form in the absence of chaperones or other protein cofactors. Anoop et al. (2014) analyzed in vitro amyloid formation of somatostatin-14 with and without disulfide bond in more detail. The proposed function is aggregation of the hormone into secretory granules as functional amyloids, which is supported by the finding that secretory granules are positive for amyloids.

In the present study, we tested a variety of CC loops for aggregation in cells rather than in vitro. Many proteins and peptides have been shown to be able to form amyloids in vitro. The hallmark of pathological or functional amyloids is that they are still able to aggregate in living cells despite the presence of chaperones, whose function is to generally prevent aggregation. We found all CC loops to have the ability to mediate ER aggregation and granule sorting, although to different extents. The differences are likely due to their intrinsic potency and/or the way they are presented by the reporter proteins, since we used the same rather short linkers.

3) The MS data in table 2 is very confusing, since half of the data points are missing. It is also not clear what the numbers in the table represent and if they are from a single experiment or multiple. As it is presented now, and as I interpret it, these results do not give support to the conclusion that CC loops form disulfide bonds. Since this is an important conclusion from the paper, these experiments need to be clarified, repeated or a different experimental approach used.

To be clearer, we have changed the way to present the data and included the results of three (in one case two) separate experiments in a new Figure 4 and changed the text accordingly.

4) As the authors state, it is well-known that the concentration of proteins in the ER will influence the ability to aggregate. In figure 1 and 2, the authors use transient overexpression to assess the ability of different CC-loops to induce aggregation in the ER. How were these results normalized to expression levels of the proteins? In later experiments the authors instead use stable cell lines expressing similar amounts of the different proteins. However, in these cells there is no obvious aggregation in the ER (see figure 4). It therefore becomes unclear what the role of ER aggregation for sorting to granules is.

The ER aggregation experiments were not normalized for expression levels. Plasmids were identical except for the short CC loop segments and produced similar transfection efficiencies. Stable cell lines with useful expression levels of CC-NPΔ could not be obtained, most likely because expression of mutant proteins inhibits growth.

To analyze granule sorting, we expressed CC fusion proteins with rapidly folding A1Pi as a reporter that does not accumulate in the ER. Stable cell lines were important to select clones with moderate and very similar expression levels.

5) What is the basal secretion of the different proteins, i.e. how much goes through the constitutive secretory pathway and how much goes through the regulated secretory pathway? The authors should show the resting secretion (before BaCl₂ addition) for all conditions tested instead of just the change in relation to control (i.e. the way data is presented now it is not possible to tell whether BaCl₂ stimulation actually cause an increased release of the peptides).

The experiment is done by comparing resting secretion (– lanes) with BaCl₂ stimulated secretion (+ lanes) in Fig. 6A and C. Stimulated secretion is calculated as a ratio of resting secretion / stimulated secretion (after normalization for cell number and

supernatant loading).

6) Lastly, the importance of CC-loops for the sorting of native peptides is unclear. The authors should test the importance of these loops for aggregation, sorting and secretion of a non-hybrid hormone with naturally occurring CC-loops (and a mutated version lacking the loop). This is important, since it is so far only shown that loops can affect the secretion of non-biologically relevant hybrid hormones.

In our previous study Beuret et al. (2017), we analyzed the segments contributing to ER aggregation of folding-incompetent mutant provasopressins and to granule sorting for folding-competent mutants of provasopressins by self-aggregation at the TGN. We found separate protein segments – vasopressin (=CCv) and the glycopeptide – to contribute to aggregation in both localizations. Our study is a follow up on the finding for vasopressin, expanding to other CC loops found in peptide hormones. Our results show that CC loops in general have the ability to aggregate and contribute to granule sorting. As exemplified by provasopressin, the CC loop may not be the only contributor. Preliminary experiments suggest the same for growth hormone. The detailed analysis of the aggregating sequences in one or more prohormone is clearly beyond the scope of our study.

****Minor comments:****

1) Stated that the 2x CC-loop constructs showed a positive effect in the cases of CCv and CCr, but this is not evaluated statistically.

Unpaired Student t-test results in p values of 0.065 for 2xCCv vs. CCv and 0.003 for 2xCCr vs. CCr. We modified the text accordingly.

2) Explain the abbreviation POMC

We have added the full name to the text.

3) Figure 6D. Paired Student's t-test is not appropriate for determining significance when data is not paired (unpaired t-tests used throughout the rest of the paper).

Only in the lubrol insolubility experiment did we find considerable shifts between experiments, which is why we used the paired t-test. However, using the unpaired t-test produces similar results. We therefore followed the reviewer's suggestion and changed the figure to show the unpaired test as in all other figures.

Reviewer #2 (Significance (Required)):

The work in this paper builds on previous work from the same group and reinforces the notion that peptide aggregation is an important part of the sorting process that controls efficient delivery of certain proteins to nascent secretory granules, and suggest that short loops formed by disulfide bridges between closely apposed cysteine residues may be part of this sorting mechanism. The paper is of general cell biological interest, but perhaps of special interest to researchers working on professional secretory cells and mechanisms of secretory protein sorting and secretion. My own research focuses on stimulus-secretion coupling pathways in secretory cells and we primarily use live cell imaging approaches to visualize different steps of secretory granule biogenesis and release.

Reviewer #3 (Evidence, reproducibility and clarity (Required)):

****Summary:****

Since the small disulfide loop of the nonapeptide vasopressin has been previously demonstrated to play a role the self-aggregation and secretory granule targeting of vasopressin precursor (Beuret et al., 2017), and as several other peptide hormones contain small disulfide loops, Reck and colleagues investigate in this study the requirement of small disulfide loops coming from four additional peptide hormones for the self-aggregation and secretory granule targeting of their precursors. Then, they studied the aggregation role of small disulfide loops in the ER and the TGN of two cell lines, COS1 and Neuro-2a. Using confocal and TEM, an aggregation has indeed been observed, although to different extents depending on the cell line. When fused to a constitutively secreted reporter protein, these disulfide loops induced their sorting into secretory granules, increased the stimulated secretion and Lubrol insolubility in endocrine AtT20 cells. All these results led the authors to hypothesize that small disulfide loops may act as a general device for peptide hormone aggregation and sorting, and therefore for secretory granule biogenesis.

****Major comments:****

The authors demonstrated the ability of small disulfide loops of peptide hormones to induce peptide precursor aggregation in ER using confocal microscopy, in COS1 and Neuro-2a cell lines, with a higher extent in COS1 cells. The authors have to moderate this conclusion and to include in their interpretation that distinct results may be due to the distinct secretory phenotype of these two cell lines: COS1 are epithelial cells, i.e. with a unique constitutive secretory pathway, while Neuro-2a as well as AtT20 cells also possess a regulated secretory pathway. Thus, the differences could be explained by the distinct molecular mechanisms involved in the formation of constitutive vesicles or secretory granules, and therefore aggregation and/or sorting processes could be distinct in the two cell types. We can also suggest to remove COS1-related results, to avoid hasty conclusions.

As suggested, we have amended the text to explain why we used COS-1 and Neuro-2a cells:

"These constructs were transiently transfected into COS-1 fibroblasts and Neuro-2a neuroblastoma cells. COS-1 cells are large and easy to analyze for aggregations by immunofluorescence microscopy, while Neuro-2a cells, although smaller and more difficult to analyze for aggregates by light microscopy, produced nice fibrillar aggregates detectable by EM for vasopressin mutants (Birk et al, 2009). Originally, Neuro-2a cells were used to study the behaviour of pathogenic diabetes insipidus mutants of vasopressin in a neuroendocrine cell type to reflect the situation in vasopressinergic neurons. For the analysis of ER aggregation, it does not matter that Neuro-2a but not COS-1 cells have a regulated secretory pathway, since the NPΔ constructs cannot reach the TGN, where regulated and constitutive pathways separate."

The data and the methods can be reproduced and the experiments are adequately replicated, using timely statistical analysis.

****Minor comments:****

- Figure 3: to complete TEM study, the concomitant use of an ER specific antibody would definitely demonstrate that small disulfide loop-containing aggregates are linked to ER compartment.

In our previous study Birk et al. (2009), we performed double-immunogold staining for vasopressin mutants and calreticulin to confirm aggregation in the ER. This anti-calreticulin antibody is unfortunately not commercially available anymore and other antibodies we tested were not suitable for immuno-EM. Instead, we colocalized PDI with CC-NPΔ constructs for immunofluorescence microscopy. Colocalization is so extensive that we believe EM confirmation to be unnecessary.

- Along abstract, introduction and discussion sections, the authors should avoid to conclude on the role of small disulfide loops on secretory granule biogenesis, but rather limit their conclusion on prohormone aggregation and targeting. Indeed, the present study did not highlight any direct molecular / physical link between disulfide loops and TGN membrane to drive secretory granule formation.

Granule biogenesis involves a number of processes including interaction of cargo components with the membrane and of the actomyosin complex with the forming buds, but also selfaggregation of cargo as functional amyloids. However, not to overstate the function of CC loops, we have reworded our statements in the Abstract to avoid the term "granule biogenesis".

Reviewer #3 (Significance (Required)):

- This study highlights small disulfide loops as novel signals for self-aggregating and secretory granule sorting of prohormone precursors in cells with a regulated secretory pathway. These results help to understand the molecular mechanism driving peptide hormone secretion, a physiological process which is crucial for interorgan communication and functional synchronization.

Moreover, their previous study revealed that vasopressin small disulfide loop is involved in toxic unfolded mutant aggregation in the ER (Beuret et al., 2017), which highlights the clinical potential of the work.

- Audience that might be interested in and influenced by the reported findings: cell biologists interested in cell trafficking, peptide hormone secretion

- My field of expertise: secretory granule biogenesis, hormone sorting, secretory cells, neurosecretion.

January 11, 2022

RE: Life Science Alliance Manuscript #LSA-2021-01279R

Prof. Martin Spiess
University of Basel
Biozentrum
Spitalstrasse 41
Basel, CH CH-4056
Switzerland

Dear Dr. Spiess,

Thank you for submitting your revised manuscript entitled "Small disulfide loops in peptide hormones mediate self-aggregation and secretory granule sorting". We would be happy to publish your paper in Life Science Alliance pending final revisions necessary to meet our formatting guidelines.

- Table should be included at the bottom of the main manuscript file or be sent as a separate file
- Table should be numbered with Arabic numerals (1, 2, 3, 4)
- there is a callout for Figure 4C in the manuscript text, but there is no such panel. Please revise
- please add callouts for Figures S1A-B and S2C to your main manuscript text

A. FINAL FILES:

B. MANUSCRIPT ORGANIZATION AND FORMATTING:

Sincerely,

January 14, 2022

RE: Life Science Alliance Manuscript #LSA-2021-01279RR

Prof. Martin Spiess
University of Basel
Biozentrum
Spitalstrasse 41
Basel, CH CH-4056
Switzerland

Dear Dr. Spiess,

Thank you for submitting your Research Article entitled "Small disulfide loops in peptide hormones mediate self-aggregation and secretory granule sorting". It is a pleasure to let you know that your manuscript is now accepted for publication in Life Science Alliance. Congratulations on this interesting work.

DISTRIBUTION OF MATERIALS:

Again, congratulations on a very nice paper. I hope you found the review process to be constructive and are pleased with how the manuscript was handled editorially. We look forward to future exciting submissions from your lab.

Sincerely,
